# Predictors of mortality among inpatients in COVID-19 treatment centers in the city of Butembo, North Kivu, Democratic Republic of Congo

**Pierre Z. Akilimali**[1,2]*, **Dynah M. Kayembe**[2], **Norbert M. Muhindo**[3,4], **Nguyen Toan Tran**[5,6]

1 Patrick Kayembe Research Center, Kinshasa School of Public Health, University of Kinshasa, Kinshasa, Congo, 2 Department of Nutrition, Kinshasa School of Public Health, University of Kinshasa, Kinshasa, Congo, 3 Assistant at the Official University of Ruwenzori in Butembo, Butembo, North Kivu, Congo, 4 Head of Manguredjipa Health Zone, Butembo, Nord Kivu, Congo, 5 Australian Centre for Public and Population Health Research, Faculty of Health, University of Technology Sydney, Sydney, NSW, Australia, 6 Faculty of Medicine, University of Geneva, Genève, Switzerland

* pierretulanefp@gmail.com

**Data Availability Statement:** Data is open in OSF: https://osf.io/qevm4 Here is the citation: Tran, N. T. (2023, December 20). Predictors of Mortality among Inpatients in COVID-19 Treatment Centers

## Abstract

Determining the risk factors for severe disease and death among hospitalized Covid-19 patients is critical to optimize health outcomes and health services efficiency, especially in resource-constrained and humanitarian settings. This study aimed to identify the predictors of mortality of Covid-19 patients in North Kivu province in the Democratic Republic of Congo. A retrospective cohort study was conducted in 6 Covid-19 treatment centers in the city of Butembo from 1 January to 31 December 2021. The time to event (death), the outcome variable, was visualized by Kaplan-Meier curves and the log-rank test was used to confirm differences in trends. Cox regression was used for all the predictors in the bivariate analysis and multivariate analysis was done using predictors found statistically significant in the bivariate analysis. The following variables were considered for inclusion to the Cox regression model: Age, Sex, Disease length, Treatment site, History of at least one co-morbidity, Body mass index, Stage according to SpO2 and the NEWS-modified score. Among the 303 participants (mean age of 53 years), the fatality rate was 33.8 deaths per 1000 patient-days. Four predictors were independently associated with inpatient death: age category ($\geq$ 60 years) (adjusted HR: 9.90; 95% CI: 2.68–36.27), presence of at least one comorbidity (adjusted HR: 11.39; 95% CI: 3.19–40.71); duration of illness of > 5 days before hospitalization (adjusted HR:1.70, 95% CI: 1.04–2.79) and peripheral capillary oxygen saturation (SpO2) < 90% (adjusted HR = 14.02, 95% CI: 2.23–88.32). In addition to advanced age, comorbidity, and length of disease before hospitalization, ambient air SpO2 measured by healthcare providers using low-tech, affordable and relatively accessible pulse oximetry could inform the care pathways of Covid-19 inpatients in resource-challenged health systems in humanitarian settings.

in the City of Butembo, North Kivu, Democratic Republic of Congo. Retrieved from osf.io/qevm4.

**Funding:** The authors received no specific funding for this work.

**Competing interests:** The authors have declared that no competing interests exist.

## I. Introduction

Corona Virus Disease 2019 (COVID-19) is an infectious disease caused by the severe acute respiratory syndrome coronavirus 2 (SARS-CoV-2). Since the discovery of the first group of cases in Wuhan City, Hubei Province, in China on December 31st, 2019, the initial COVID-19 outbreak has become one of the most significant global health threats.

It was declared a public health emergency of international concern on January 30th, 2020, by the World Health Organization (WHO) and a pandemic on March 11th, 2020 [1, 2]. By 2021, COVID-19 had affected almost all countries around the world.

According to the WHO situation reports, as of October 4th, 2022, the number of confirmed COVID-19 cases reached more than 615,777,700 cases and 6,527,192 deaths (1.06% case- fatality rate) [3]. In the African region, the first case of COVID-19 was reported on February 14th, 2020, in Egypt and on February 17th, 2020 in sub-Saharan Africa (Nigeria). Since then, all 54 African states have been affected. As of September 21st, 2021, the WHO African Region has recorded 8,166,634 cases and 206,740 deaths (2.53% case-facility rate) [4–7]. The Democratic Republic of Congo (DRC) recorded its first case on March 10th, 2020. Since then, the cumulative number of cases up to September 3rd, 2022 was 92,942 cases, of which 92,940 were confirmed, with 1,357 deaths, representing a case-fatality rate of 1.5% [8]. According to the epidemiological report in North Kivu, as of may 22nd, 2022, the province recorded 10,049 cases, of which 9,144 recovered (91.0%) and about 600 died (5.9%). These figures place the province in second place after the provincial city of Kinshasa in terms of disease burden [9]. On the same date, the report indicates that the city of Butembo, with its two health zones, had 1280 cases, of which 1157 (90.4%) recovered and 116 (9.1%) died. These two health zones concentrated 20% of the province's mortality [9].

The signs and symptoms of COVID-19 vary considerably, with clinical features ranging from asymptomatic presentation to fatal respiratory distress and multiorgan failure [10, 11]. COVID-19 does not exert an equal impact on the different affected regions in the world, with a wide variation in the reported proportion of patients with severe disease and death [12, 13]. Morbidity accounts for 6% of cases in the Eastern Mediterranean region, 30% in Europe, 39% in the American region, 19% in South Asia, 3% in the Pacific and 4% in Africa [4, 14]. The consequences are palpable worldwide at all levels with significant economic, social, health and financial costs [15].

Strategies to prevent COVID-19 transmission have included border closures, the creation of quarantine centers, self-isolation, enforcement of containment measures, surveillance, routine screening, and, recently, vaccination, with the first vaccine deliveries to Africa in March 2021 [15].

Studies were done worldwide to determine COVID-19 risk factors and patient characteristics and recently the impact of vaccination on hospitalization [1, 2, 6, 15]. Overall, risk factors for severe disease and death included advanced age, male sex, history of comorbidity, poor nutritional status, a severe form of the disease on admission, vaccination status, delay in admission to intensive care, etc. To our knowledge, by the time we conducted our research in 2021, such a study had not yet taken place in the region. Therefore, our study proposed to identify the predictors of mortality of COVID-19 patients who were hospitalized in the city of Butembo from January 1st to 31 December 31st, 2021.

## II. Methods

This study was conducted in six COVID-19 treatment centers (CTC) in the Katwa and Butembo health zones of the city of Butembo, North Kivu province, Democratic Republic of Congo. The CTCs were located at the Hospital of Women Committed to the Promotion of

Integral Health, the Kitatumba Referral General Hospital, the Graben University Clinics, the Katwa Referral General Hospital, the Matanda Hospital and the Ngote Hospital.

A retrospective cohort study was conducted from 1 January to 31 December 2021.The study population included all patients hospitalized in one of the CTCs in the city of Butembo in 2021 with a positive reverse transcriptase-polymerase chain reaction result (RT-PCR +). The statistical units were the COVID-19 patients registered in each health center. We excluded patients without investigation forms or whose clinical records could not be found. In total, 303 Covid-19 patient records were found. Delay between symptom onset and hospital admission was the exposure variable. Patients were divided into two groups: 188 patients hospitalized for ≤ 5 days (Short delay) and 115 patients hospitalized for> 5 days (long delay). Data was extracted by the authors who are all medical doctors.

## Variables

The time to event (death) was the outcome variable. Predictor variables were age, sex, occupation, clinical stage of disease based on signs and symptoms, ambient air oxygen saturation (SpO2), respiratory rate, NEWS-modified score (National Early Warning Score), history of comorbidities, number of comorbidities, types of comorbidities, body mass index (BMI), duration of disease before hospitalization, date of discharge or last news about the patient. NEWS comprises seven parameters: respiration rate, SpO2, any supplemental oxygen, temperature, systolic pressure, heart rate, level of consciousness (Table 1). It allows the classification of COVID-19 patients into three severity categories: low (aggregate 1–4), medium (aggregate 5–6), and high (aggregate 7 or more). Because some parameters (blood pressure, supplemental oxygen) in the patient's records were missing, we modified the scoring of some parameters to obtain the NEWS-modified score with five parameters. The time variable was calculated from the date of enrolment (admission at the hospital) to the end point (death, discharge or end of the study).

## Data collection

An Excel database was used to collect data from the patient records, registers and CTC databases. After verification and validation, it was imported into SPSS (Statistical Package for the Social Sciences) version 26 and Stata 17 (StataCorp, College Station, TX) for analysis. Before the actual analysis, we transformed the variables of age, BMI, respiratory rate, ambient air oxygen saturation, number of comorbidities, and Hb level into categorical variables.

## Statistical analysis

Descriptive statistics were used to describe the basic characteristics of the study data. For continuous variables, means and standard deviation (SD) were calculated for normally distributed

**Table 1. Score NEWS–adapted from the NEWS2 scoring matrix.**

| Vital signs | 3 | 2 | 1 | 0 | 1 | 2 | 3 |
|---|---|---|---|---|---|---|---|
| Temperature | Hypothermia | | Fever | No fever | | | |
| Heart rate | | | | Normal | | | Abnormal |
| Respiratory rate | ≤ 8 | | 9–11 | 12–20 | | 21–24 | ≥ 25 |
| Ambient air SpO2 | ≤ 91 | 92–93 | 94–95 | ≥ 96 | | | |
| Consciousness level | | | | Lucid/alert | | | Confused |

Classification: mild or moderate (NEWS score < 5); severe (NEWS score ≥ 5) [16]

continuous variables and median with interquartile range (IQR) for non-normally distributed continuous variables; for categorical data, proportions and their respective 95% confidence intervals were calculated. We used the chi-square test and Fisher´s exact test when appropriate.

We calculated proportions, where the main outcome variable was death. We determined the incidence rate of recorded death events per 1000 patient-day (p-d) from the date of enrollment. The survival probabilities of participants according to the predictor variables were visualized by Kaplan-Meier curves and the log-rank test was used to confirm differences in trends. Cox regression was used for all the predictors in the bivariate analysis and multivariable analysis was done using predictors found statistically significant in the bivariate analysis. The following variables were considered for inclusion to the Cox regression model: Age, Sex, Disease length, Treatment site, History of at least one co-morbidity, Body mass index, Stage according to SpO2 and the NEWS-modified score. The interaction age and SpO2 was not significant and was not included in the model. The interaction between the comorbidity and «the delay between symptom onset and hospital admission» was not significant and was not included in the model. The interaction between age and «the delay between symptom onset and hospital admission» was not significant and was not included in the model. The interaction age and comorbidity was significant and was included in the model. We then compared the model with the interaction to the model without the interaction using the lrtest command. The significant lrtest indicated that we reject the null hypothesis that the two models fit the data equally well and concluded that the bigger model with the interaction fits the data better than the smaller model which did not include the interaction.

The proportionality test based on Schoenfeld residuals verified compliance with the assumption of the proportionality of risks (refer to S1 Table). The Test of proportional hazards shown that the assumption was not violated as presented in S1 Fig. Regarding the goodness of fit of the final model, we have seen that the hazard function follows the 45-degree line very closely except for very large values of time. Overall, we would conclude that the final model fits the data very well (refer S2 Fig). We assessed multicollinearity using variance inflation factors (VIFs) greater than 2.1. The proportionality test based on Schoenfeld residuals verified compliance with the assumption of the proportionality of risks. All tests were two-tailed with 95% confidence intervals and considered statistically significant when p-value < 0.05. Dataset can be found in osf:

https://osf.io/qevm4/?view_only=9da898c21cb94c0385e3c0d5342b283d

## Ethical considerations

Prior authorizations were obtained from the health authorities in North Kivu province (Head of the Health Division, Coordinator of the DPS/Butembo branch, the Head of Katwa and Butembo Health Zone and the Directors of the selected hospitals). To ensure confidentiality, we deidentified the variables in the database. With regard to informed consent, we did not have any contact with the patients, so no biological procedures were used in the collection or processing of the data. The use of the results of this study will be limited to the strict exploitation related to its objectives and the authors have reported no conflict of interest. The protocol for this study had received ethical approval from the School of Health Ethics Committee (reference number: ESP/CE/138/2021).

All the study sites have had a blanket data policy informing their patients of the following: "For all rare and emerging diseases for which science still needs further data, hospitalized patients have accepted that their data be used for research in order to improve treatment." The Kinshasa School of Public Health Ethics Committee approved the study.

## III. Results

There were 303 participants who had a mean age of 53 ± 22 years. Those who consulted within 5 days of symptom onset were younger than those who consulted ≥ 5 days (51 vs. 57 years; p = 0.014). Overall, 43% were at least 60 years old, 46% were male, and 6.4% were healthcare professionals. The distribution of patients in the population from which the two groups were drawn was identical according to sociodemographic characteristics, except for age and site (Matanda and Katwa particularly) (Table 2).

Of 303 patients included, 45.9% had at least one comorbidity of whom 25.2% had at least two comorbidities. The most frequent comorbidities were hypertension 18.8% (57/303) and diabetes 17.8% (54/303). The distribution of patients in the ≤ 5 days (short) and > 5 days (long) hospitalization groups was similar in terms of comorbidities (Table 2).

Cough and asthenia were present in more than 80% of cases, anorexia, headache and dyspnea in 50–79% of cases, and fever, digestive, and taste disorders and other signs in less than 50% of cases. The distribution of patients between the short and long-hospitalization groups differed in terms of symptoms, except for cough, headache, loss of taste, and digestive and cardiac disorders (Table 2).

The mean oxygen saturation on admission was 85.60% ± 12.72% (81.91 ± 14.29 in the exposed (> 5 days (long) hospitalization group) and 87.85 ± 11.11 in the unexposed (in the ≤ 5 days (short) hospitalization group)). Patients admitted early had higher oxygen saturation than those who visited late. Patients with a severe form accounted for 61% of all participants based on the m-NEWS score, 49% based on their SpO2 level and 55% based on symptoms at admission. Regardless of the classification used, the distribution of patients between the short and long-hospitalization groups differed according to disease severity upon admission (Table 3).

The fatality rate was 33.8 deaths per 1000 patient-days. The rate was 46 deaths per 1000 patient-days in the exposed group (> 5 days (long) hospitalization group) versus 25.6 deaths per 1000 patient days in the unexposed group (in the ≤ 5 days (short) hospitalization group) (Table 4).

After adjustment, four predictors were independently associated with inpatient death: older patients (≥ 60 years) had a higher risk of death than the reference group(<40 years) (adjusted HR: 9.90; 95% CI: 2.68–36.27) (see Fig 1 and Table 4), having at least one comorbidity had a higher risk of death than patients without comorbidity (aHR: 11.39; 95% CI: 3.19–40.71) (see Fig 2 and Table 4); patient admitted after "long delay" between symptom onset and hospital admission duration of illness of > 5 days before hospitalization had a higher risk of death than patients admitted after "short delay" (aHR:1.70, 95% CI: 1.04–2.79) and Patients with SpO2 < 90% had a higher risk of death than patients with SpO2 ≥ 95% (HRa = 14.02, 95% CI: 2.23–88.32) (Table 4). The interaction age and comorbidity was significant and was included in the model.

## IV. Discussion

This study focused on the predictors of survival of Covid-19 patients hospitalized from 1 January to 31 December 2021 in the city of Butembo. Advanced age above 60 years, low free air SpO2 on admission, history of comorbidity and late admission (> 5 days after symptom onset) were predictors of mortality. Overall, these predictors have been previously reported in studies conducted in China, Europe, the USA and other African countries [2, 4, 10, 17–27]. Another new predictor introduced in the present study was the duration of illness > 5 days before hospitalization. Knowledge of these factors is an important advocacy tool for covid-19 patients in humanitarian and resource-limited settings. The use of NEWS-modified or original NEWS score, the measurement of oxygen saturation and search for co-morbidities must be

**Table 2. Socio-demographic characteristics and clinical signs of Covid-19 patients on admission to the Covid-19 treatment centers in the city of Butembo from January 1st, to December 31st, 2021.**

| Characteristics on admission | All cases | | Delay between symptom onset and hospital admission | | | | |
|---|---|---|---|---|---|---|---|
| | | | > 5 days | | ≤ 5 days | | p |
| | n = 303 | % | n = 115 | % | n = 188 | % | |
| Age (median, IQR) | 55.0(40–71) | | 53.0 (39–68) | | 60.5 (40–75) | | **0.032*** |
| Age(range) | | | | | | | 0.139 |
| < 40 years | 83 | 27.4 | 27 | 23.5 | 56 | 29.8 | |
| 40–59 years | 89 | 29.4 | 30 | 26.1 | 59 | 31.4 | |
| ≥ 60 years | 131 | 43.2 | 58 | 50.4 | 73 | 38.8 | |
| Gender | | | | | | | 0.313 |
| Male | 139 | 45.9 | 57 | 49.6 | 82 | 43.6 | |
| Female | 164 | 54.1 | 58 | 50.4 | 106 | 56.4 | |
| Profession* | n = 234 | | n = 90 | | n = 144 | | 0.601 |
| Cultivator | 72 | 30.8 | 32 | 35.6 | 40 | 27.8 | |
| Housekeeper | 50 | 21.4 | 19 | 21.1 | 31 | 21.5 | |
| Shopkeeper | 30 | 12.8 | 10 | 11.1 | 20 | 13.9 | |
| Health care professional | 15 | 6.4 | 7 | 7.8 | 8 | 5.6 | |
| None and other professions | 67 | 28.6 | 22 | 24.4 | 45 | 31.3 | |
| Signs and symptoms | | | | | | | |
| Fever on admission | 140 | 46.2 | 38 | 33.0 | 102 | 54.3 | **<0.001** |
| Dry or wet cough | 286 | 94.4 | 109 | 94.8 | 177 | 94.1 | 0.816 |
| Dyspnea | 164 | 54.1 | 78 | 67.8 | 86 | 45.7 | **<0.001** |
| Shortness of breath | 247 | 81.5 | 107 | 93.0 | 140 | 74.5 | **<0.001** |
| Asthenia | 180 | 59.4 | 63 | 54.8 | 117 | 62.2 | 0.2 |
| Headache | 163 | 53.8 | 80 | 69.6 | 83 | 44.1 | **<0.001** |
| Anorexia | 23 | 7.6 | 8 | 7.0 | 15 | 8.0 | 0.744 |
| Loss of taste | 41 | 13.5 | 18 | 15.7 | 23 | 12.2 | 0.399 |
| Digestive disorders | 11 | 3.6 | 5 | 4.3 | 6 | 3.2 | 0.605 |
| Heart rhythm disorders | 17 | 5.6 | 11 | 9.6 | 6 | 3.2 | 0.019 |
| Altered consciousness | 4 | 1.3 | 2 | 1.7 | 2 | 1.1 | 0.979 |
| HIV positive | 57 | 18.8 | 23 | 20 | 34 | 18.1 | 0.679 |
| Hypertension | 54 | 17.8 | 23 | 20 | 31 | 16.5 | 0.438 |
| Diabetes | 6 | 2 | 4 | 3.5 | 2 | 1.1 | 0.299 |
| Obstructive lung disease | 6 | 2 | 3 | 2.6 | 3 | 1.6 | 0.825 |
| Obesity | 139 | 45.9 | 58 | 50.4 | 81 | 43.1 | 0.213 |
| Sites | | | | | | | **<0.001** |
| Pepsi | 19 | 6.3 | 7 | 6.1 | 12 | 6.4 | |
| Kitatumba | 42 | 13.9 | 15 | 13.0 | 27 | 14.4 | |
| CUG | 52 | 17.2 | 33 | 28.7 | 19 | 10.1 | |
| Katwa | 75 | 24.8 | 17 | 14.8 | 58 | 30.9 | |
| Matanda | 60 | 19.8 | 28 | 24.3 | 32 | 17.0 | |
| Ngote | 55 | 18.2 | 15 | 13.0 | 40 | 21.3 | |

CUG: Cliniques Universitaires du Graben (Graben University Clinics); *Median test

systematic. These parameters may help to select the most at-risk patients with a poor prognosis and to intensify surveillance. In almost all hospitals in Africa in general, and in the DRC in particular, the measurement of oxygen saturation was not systematically used of times due to the absence of an oximeter.

**Table 3. Clinical stage of inpatients on admission to Covid-19 treatment centers in the city of Butembo from January 1 to December 31, 2021.**

| Stade de la maladie à l'admission | | | Delay between symptom onset and hospital admission | | | | p-value |
|---|---|---|---|---|---|---|---|
| | Number of cases | | > 5 days | | ≤ 5 days | | |
| | (n) | % | (n) | % | (n) | % | |
| SpO2 (mean, SD) | 85.60 ± 12.7 | | 81.91 ± 14.3 | | 87.85 ± 11.1 | | <0.001 |
| Stage according to SpO2 | | | | | | | 0.001 |
| Mild | 61 | 20.1 | 12 | 10.4 | 49 | 26.1 | |
| Moderate | 94 | 31.1 | 34 | 29.6 | 60 | 31.9 | |
| Severe | 148 | 48.8 | 69 | 60.0 | 79 | 42.0 | |
| Modified national early warning score | | | | | | | <0.001 |
| < 5 | 119 | 39.3 | 29 | 25.2 | 90 | 47.9 | |
| ≥ 5 | 184 | 60.7 | 86 | 74.8 | 98 | 52.1 | |
| Disease severity on admission | | | | | | | 0.001 |
| Mild | 23 | 7.6 | 5 | 4.3 | 18 | 9.6 | |
| Moderate | 114 | 37.6 | 31 | 27.0 | 83 | 44.1 | |
| Severe | 166 | 54.8 | 79 | 68.7 | 87 | 46.3 | |
| Total | 303 | 100 | 115 | 100 | 188 | 100 | |

Our study found that the risk of death increases with age. Compared to patients under 40 years of age, the risk of death is four times higher in patients > 60 years. It is well established that older age groups are more likely to have comorbidities and are more susceptible to acquire SARS-CoV-2, severe forms of the disease, and higher mortality risk than younger patients [28]. The difference between the age groups lies in the ability to fight the infection [29]. Several studies have proposed multiple pathogenic mechanisms for severe disease in the elderly population, including low levels of angiotensin-converting enzyme 2 (ACE2) in the elderly [30], age-related difficulty in clearing particulate matter from small airways [31], excessive release of inflammatory mediators in the elderly ("inflammaging") [32]. This finding is similar to recent retrospective studies from Egypt [33], Iran [27] and Pakistan [34] and Kinshasa [35–38]. This is also consistent with other reports from outside the Eurogio Meuse-Rhine (EMR) countries [39, 40].

Our study also revealed that the risk of death for patients with a history of at least one comorbidity (the most common being hypertension and diabetes) is more than double. Indeed, the high expression of ACE2 receptors in highly differentiated airway epithelial cells underlies susceptibility to SARS-CoV-2. When hypertensive patients are infected with the SARS-CoV2, blood pressure regulation becomes more complicated and difficult to control, and the cardiovascular risk is amplified [27, 41]. Diabetic patients are more likely to have severe Covid-19 complications because their high blood glucose levels promote viral growth and impair their immune function and ability to resist infection, setting the stage for secondary bacterial and viral co-infections. In addition, if diabetes complications occur, the risk of multiorgan failure and death is significantly increased. One study reported that mortality and multiorgan injury were significantly higher in Covid-19 patients with type-2 diabetes than in patients without diabetes (HR, 1.5). Such an association has been demonstrated by numerous studies worldwide, reminding us of the need to control chronic non-communicable diseases and pay special attention to people with comorbidities during epidemics [23, 26, 27, 41–50].

Regarding SpO2 at admission, our study demonstrated the risk of death for patients admitted with SpO2 < 90%, which is more than 13 times compared to patients with normal saturation (≥95%). Low ambient air SpO2 is strongly associated with poor outcomes on admission to the hospital and, therefore, is part of the classification of disease severity [47, 51]. Indeed,

**Table 4. Predictors of mortality of Covid-19 inpatients in the city of Butembo from January 1 to December 31, 2021.**

| Variables | (n) | Death | patient-day | Death incidence /1000 patient-day | aHR | 95% CI | p-value |
|---|---|---|---|---|---|---|---|
| Delay between symptom onset and hospital admission | | | | | | | |
| ≤ 5 days | 188 | 37 | 1507 | 24,6 | 1 | | |
| > 5 days | 115 | 45 | 1001 | 45,0 | 1.70 | 1.04–2.79 | 0.035 |
| Age (years) | | | | | | | |
| < 60 | 172 | 26 | 1465 | 17,7 | 1 | | |
| ≥ 60 | 131 | 56 | 1042 | 53,7 | 9.9 | 2.68–36.27 | 0.001 |
| Gender | | | | | | | |
| Female | 164 | 41 | 1320 | 31,1 | 1 | | |
| Male | 139 | 41 | 1188 | 34,5 | 1.39 | 0.86–2.24 | 0.182 |
| Treatment site | | | | | | | |
| FEPSI | 19 | 3 | 212 | 14,2 | 1 | | |
| KITATUMBA | 42 | 12 | 326 | 36,8 | 1.62 | 0.34–5.04 | 0.400 |
| CUG | 52 | 13 | 648 | 20,1 | 0.27 | 0.05–0.79 | 0.022 |
| KATWA | 75 | 18 | 574 | 31,4 | 0.69 | 0.16–2.15 | 0.425 |
| MATANDA | 60 | 29 | 422 | 68,7 | 1.31 | 0.30–3.64 | 0.950 |
| NGOTE | 55 | 7 | 326 | 21,5 | 0.61 | 0.12–2.18 | 0.367 |
| History of at least one co-morbidity | | | | | | | |
| No | 164 | 20 | 1361 | 14,7 | 1 | | |
| Yes | 139 | 62 | 1147 | 54,1 | 11.39 | 3.19–40.71 | < 0.001 |
| Stage according to SpO2 | | | | | | | |
| Mild | 61 | 2 | 487 | 4,1 | 1 | | |
| Moderate | 94 | 10 | 738 | 13,6 | 3.56 | 0.71–17.82 | 0.122 |
| Severe | 148 | 70 | 1283 | 54,6 | 14.02 | 2.23–88.32 | 0.005 |
| m-NEWS | | | | | | | |
| < 5 | 119 | 9 | 1088 | 8,3 | 1 | | |
| ≥ 5 | 184 | 73 | 1420 | 51,4 | 0.72 | 0.22–2.27 | 0.571 |
| Age(≥ 60)#History of at least one co-morbidity | | | | | | 0,15 | 0.04–0.61 | 0.008 |
| Total | 303 | 82 | 2508 | 32,7 | | | |

CUG: Cliniques Universitaires du Graben (Graben University Clinics)

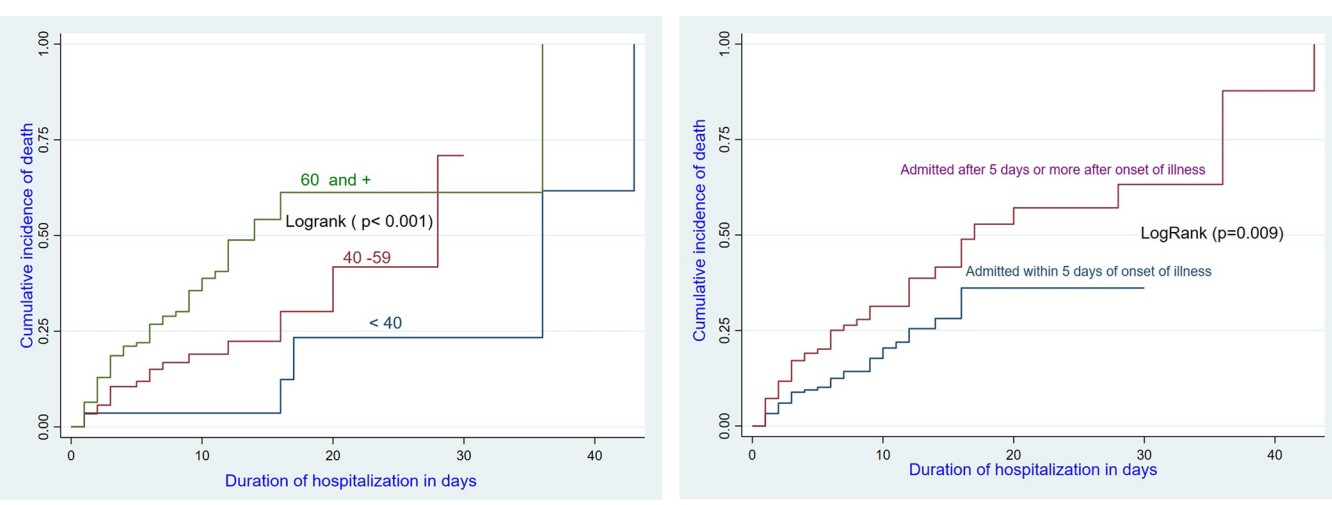

**Fig 1. Survival of hospitalized COVID-19 patients by age and time from disease onset to admission.**

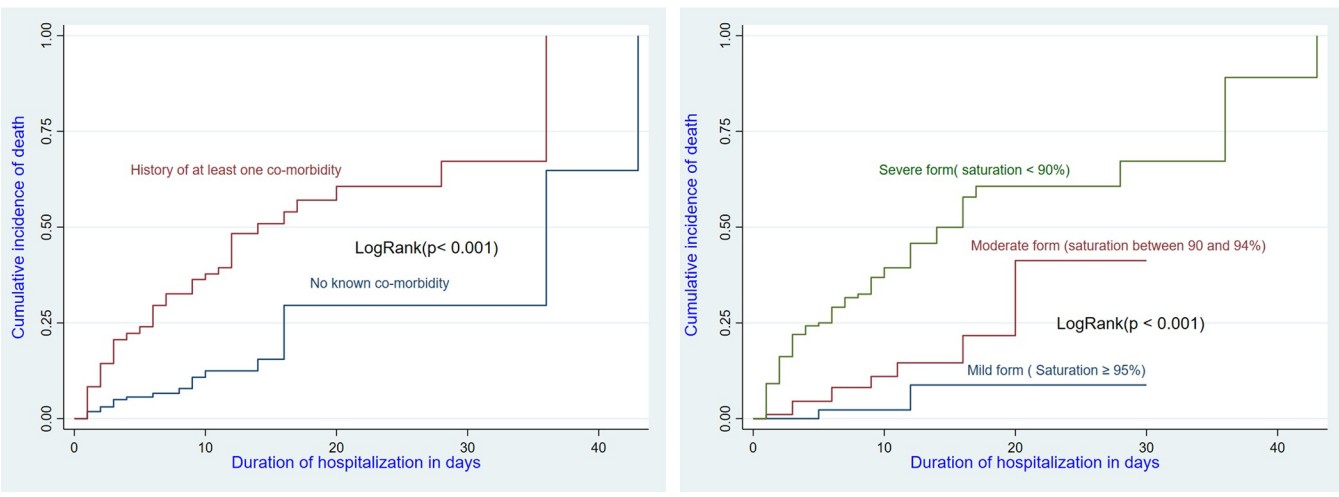

**Fig 2. Survival of hospitalized COVID-19 patients by presence of comorbidity and disease stage (based on saturation) at admission.**

the degree of ambient air O2 saturation reflects the level of the respiratory system function [38]. Such observation has been made by other authors and reiterates the need to take into account SpO2 levels in the prioritization of patients for admission to intensive care units [35, 38, 39, 44]. The wide confidence interval for SpO2 is due to the small number of events in the reference group, we have only 2 cases of death in mild severe patients.

In our study, delay in consulting was significantly and independently associated with mortality, with a risk of more than doubling. Indeed, in all infectious diseases, the duration of symptoms before hospitalization has an impact on the outcome. In fact, delayed onset of supportive care affects clinical outcomes by enabling immuno-inflammatory and thrombotic responses [44]. This result mirrors the findings of other studies, such as that of Wen-Hua Liang et al. in China [52].

On the other hand, our study did not establish a significant association between gender and death as in the study of Cummings MJ et al. [44]. Similarly, Nasiri et al. showed that there was no significant difference between men and women in terms of admissions to intensive care units [53]. Another Egyptian study of 260 patients with COVID-19 showed no significant association between male gender and risk of severe disease [54]. In Italy, Ciceri et al. reported no significant association between the female gender and the risk of severe disease [55]. Other reports from the Eurogio Meuse-Rhine countries demonstrated similar findings [27, 33, 37]. Although recent surveillance has shown that male patients were more likely to acquire severe infection and die with COVID-19 [56], This sex-specific difference was observed in previous SARS outbreaks [57].

The interpretation of the results of the present study should consider some of its limitations. First, although we tried to be as complete as possible, the retrospective nature of this study limited the availability of data due to issues of archiving records, quality of the data found in the patient records, and possible errors in assessing patient parameters at admission. Second, record incompleteness, and the limited capacity in diagnosing other comorbidities could have biased the outcome as we could not input more predictors that could have explained mortality. The fact that we did not have the opportunity to analyze the impact of COVID vaccination in our study is another limitation. Regarding COVID-19 vaccines, the roll-out started on April 19th, 2021. However, the uptake was slow initially, by the time of data collection, only about 7% of the target population (entire population >18) had received at least one dose in the entire

country. This low coverage may affect case fatality. This may have also substantial impact on study results as not all cases may have ended up to the hospital thereby creating reporting bias of the outcome.

Finally, because the date of symptom onset is based on self-report, a bias related to the patient's recollection may have misclassified patients at admission. However, to reduce the risk of error, we linked the different sources of data (patient and investigation forms, registers, and databases). We ensured the streamlining of clinical assessment and equipment used to collect patient parameters upon admission in each CTC.

Despite these limitations, this study has nevertheless the merit of having shown a new factor in the arsenal of predictors of survival of patients hospitalized for COVID-19 in the city of Butembo from January to December 2021. These results can also be used to prioritize actions according to the characteristics of the population and the triage of patients upon admission to the health facilities in the city of Butembo and in similar resource-limited contexts.

## V. Conclusions

Advanced age, history of comorbidity, low level of ambient air SpO2 at admission and long duration of the disease before hospitalization were predictors of survival of patients hospitalized for COVID-19 in Butembo. Therefore, the following actions are recommended: (a) health policymakers and authorities must strengthen barrier measures and make them mandatory to protect the elderly and those with a history of comorbidity; (b) health authorities, along with technical and financial partners, should incentivize symptomatic patients to seek care at dedicated health structures as early as possible; and (c) health care providers should measure ambient air SpO2 as part of criteria for admission to intensive care units.

## Supporting information

**S1 Table. Test of proportional hazards assumption.**
(DOCX)

**S1 Fig. Test of proportional hazards assumption by predictor.**
(ZIP)

**S2 Fig. Goodness of fit of the final model.**
(TIF)

## Author Contributions

**Conceptualization:** Pierre Z. Akilimali, Dynah M. Kayembe, Nguyen Toan Tran.

**Data curation:** Pierre Z. Akilimali, Dynah M. Kayembe, Norbert M. Muhindo.

**Formal analysis:** Pierre Z. Akilimali, Norbert M. Muhindo.

**Funding acquisition:** Norbert M. Muhindo.

**Investigation:** Pierre Z. Akilimali, Norbert M. Muhindo.

**Methodology:** Pierre Z. Akilimali, Dynah M. Kayembe, Norbert M. Muhindo, Nguyen Toan Tran.

**Project administration:** Norbert M. Muhindo.

**Resources:** Pierre Z. Akilimali, Norbert M. Muhindo.

**Software:** Pierre Z. Akilimali, Norbert M. Muhindo.

**Supervision:** Pierre Z. Akilimali, Norbert M. Muhindo.

**Validation:** Pierre Z. Akilimali.

**Visualization:** Pierre Z. Akilimali.

**Writing – original draft:** Pierre Z. Akilimali, Dynah M. Kayembe, Norbert M. Muhindo, Nguyen Toan Tran.

**Writing – review & editing:** Pierre Z. Akilimali, Dynah M. Kayembe, Norbert M. Muhindo, Nguyen Toan Tran.

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
