## [Decision Letter · Decision Letter 0]

5 Apr 2023

PGPH-D-23-00180

Predictors of Survival among Inpatients in COVID-19 Treatment Centers in the City of Butembo, North Kivu, Democratic Republic of Congo

Dear Dr. Akilimali,

Thank you for submitting your manuscript to PLOS Global Public Health. After careful consideration, we feel that it has merit but does not fully meet PLOS Global Public Health’s publication criteria as it currently stands. Therefore, we invite you to submit a revised version of the manuscript that addresses the points raised during the review process.

Two independent reviewers have assessed the manuscript. Although they found the work interesting, they have raised some concerns that should be carefully addressed. We consider this a major revision. Please note that major revisions may be subject to re-review.

We look forward to receiving your revised manuscript.

Kind regards,

Chrispin Chaguza, Ph.D

Academic Editor

Journal Requirements:

1. In the online submission form, you indicated that your data will be submitted to a repository upon acceptance.  We strongly recommend all authors deposit their data before acceptance, as the process can be lengthy and hold up publication timelines. Please note that, though access restrictions are acceptable now, your entire data will need to be made freely accessible if your manuscript is accepted for publication. This policy applies to all data except where public deposition would breach compliance with the protocol approved by your research ethics board. If you are unable to adhere to our open data policy, please kindly revise your statement to explain your reasoning and we will seek the editor's input on an exemption. Please be assured that, once you have provided your new statement, the assessment of your exemption will not hold up the peer review process.

2. Please send a completed 'Competing Interests' statement, including any COIs declared by your co-authors. If you have no competing interests to declare, please state "The authors have declared that no competing interests exist". Otherwise please declare all competing interests beginning with the statement "I have read the journal's policy and the authors of this manuscript have the following competing interests:"

3. If you did not receive any funding for this study, please simply state: “The authors received no specific funding for this work.”"

Reviewers' comments:

Reviewer's Responses to Questions

**Comments to the Author**

1. Does this manuscript meet PLOS Global Public Health’s publication criteria? Is the manuscript technically sound, and do the data support the conclusions? The manuscript must describe methodologically and ethically rigorous research with conclusions that are appropriately drawn based on the data presented.

Reviewer #1: Partly

Reviewer #2: Partly

2. Has the statistical analysis been performed appropriately and rigorously?

Reviewer #1: Yes

Reviewer #2: No

3. Have the authors made all data underlying the findings in their manuscript fully available (please refer to the Data Availability Statement at the start of the manuscript PDF file)?

Reviewer #1: No

Reviewer #2: Yes

4. Is the manuscript presented in an intelligible fashion and written in standard English?

Reviewer #1: Yes

Reviewer #2: No

5. Review Comments to the Author

**Reviewer #1: Manuscript Summary**

This manuscript identifies predictors of COVID-19 mortality among hospitalized patients in the Democratic Republic of Congo. The authors use cox regression to identify factors that are associated with high mortality rate accounting for time-dependence observed outcome (death). They show that mortality risk is likely influenced by older age, presence of comorbidity, self-defined duration of illness before hospitalization, and ambient air SpO2. Risk factors for severe COVID-19 as well as mortality from COVID-19 have widely been discussed and this study does not add new evidence to the discussion. However, authors claim their study was undertaken at the time when other studies had not yet published their work. On the positive outlook, there are limited studies in Africa describing COVID-19 experiences in hospital settings and this study may be relevant for making a case for experiences in resource-constrained and a humanitarian setting. There are major revisions that need to be made for scientific soundness. Some statistical analysis consideration have been suggested. And lastly, ethics statement is not very clear.

Since line numbering was not put in the article I will describe where suggested changes may go;

Title

1.    The authors should modify the title to “Predictors of mortality among ….” And not “Predictors of survival among ….” because that is what they present in the manuscript. The authors should also make changes on every occasion where this occurs in main text.

Abstract

2.    In the abstract, SpO2 should be written in full before abbreviation

3.    I think the appropriate epidemiologic study design is retrospective cohort study or prospective cohort study as was not clear but not analytic cohort study.

4.    All considered predictors should be listed in the Methods section so can have assess what went into the models

5.    The wide confidence interval for SpO2 suggests few data points considered, which may likely cause imbalance in model estimation, and authors could ensure to refit models without this variable to assess the impact of this variable

Introduction

6.    Paragraph 5: The word reported must precede these statements as undocumented patients are likely more than the presented numbers.

a.    with a wide variation in the **reported** proportion of patients with severe disease and death

b.    **Reported** morbidity accounts for 6% of cases in the Eastern Mediterranean region.

7.    When was Covid 19 vaccine introduced in this setting? And if at this time it was already introduced, were some patients vaccinated and others not prior to hospitalization? May be good to discuss COVID-19 vaccination situation in this setting.

Methods

8.    Paragraph 1: Could authors discuss the representation of these hospitals with respect to quality standard of care compared to other hospitals in the country.

9.    Paragraph 3: Could authors discuss what proportion of records were excluded. Did this introduce selection bias? and potentially affecting the validity of results?

10.    Paragraph 3: Why 5 days categorization? My understanding is that authors are introducing an exposure of interest variable based on patient self-reported time when symptoms may have commenced before being hospitalized. Then, why did the authors choose to split this duration by 5 days threshold? Why is this duration important? There is need for some justification of choices made.

11.    Data collection: Could authors also justify thresholds used to transform continuous variables into categorical variables

12.    Statistical analysis: All tests were two-tailed with 95% confidence intervals and considered **statistically** significant when p-value < 0.05.

Ethical considerations

13.    The first and last sentences in the first paragraph are irrelevant. Also, the last paragraph is irrelevant. Additionally, I think this section should clearly state if patients consented to participate in the study, and which ethics committee approved the study and the approval number, which in my opinion is not very clear.

Results

14.    All presented results do not have referenced table or figure where the readers can refer for further details. Could authors cite relevant table or figure for each paragraph.

15.    Paragraph 2: “Regarding comorbidity, 45.9% had at least one and 25.2% (35/139) had at least two” could authors consider rephrasing this sentence as “Of 303 patients with comorbidities, 45.9% had at least one of whom 25.2% had at least two comorbidities.” As it is it’s a bit confusing especially when denominators are not clearly shown.

16.    Paragraph 2: Is it possible to quantify the distribution and write the results in text rather than making descriptive sentence. e.g., <=5d mean X SDX or Median X IQR X-X etc. If the same could be done in all places where the distributions are simply described.

17.    Last paragraph: Could author include all reference groups for the estimated predictor level rather than leaving readers guessing what the hazard ratios comparisons are with.

18.    Table 3: Some percentages are not adding to 100%,  and all percentage could be kept to 1 decimal place for consistence.

19.    Table 4: Not sure why authors have been calling the exposure of interest different names e.g., Disease length, Disease duration before hospitalization, Time from symptom to hospitalization. It would be good to be consistent with exposure variable name. Also, I guess you meant to bold its p value for being statistically significant

20.    In all the plots, the y-axis says: “Cumulative incidence of death” but the y-axis seems to suggest it’s probability and not necessarily incidence (as units are not present and it’s restricted between 0 and 1). It should rather be named “Mortality probability” or if the inverse is applied with curves starting from 1 on downward trend, the y-axis should be named “Survival probability”. I would think the latter is frequently used.

21.    It’s also relevant to include confidence intervals in the survival curves to have a clearly understanding of uncertainty, and if authors are happy to do so.

Discussion

22.    Paragraph 1: The first paragraph of the Discussion should also discuss the "So what question" e.g., what’s the implication of these key findings or what does it mean now that these factors are identified?

23.    Paragraph 2: What does it mean, older age groups are more susceptible to infection e.g., is it to acquire SARS-CoV-2 or it is about invasiveness? Could the authors clarify.

24.    As mentioned beforehand, COVID-19 vaccines have had substantial impact on mortality from COVID-19 disease. A paragraph should be included to discuss the impact of these vaccines on the results.

25.    Authors should also discuss the implication of undocumented COVID-19 deaths in DRC due to resources-constraints and how this could affect their results in this manuscript.

**Reviewer #2: Manuscript summary**

The paper aims at identifying the predictors of survival for Covid-19 patients in North Kivu province in the Democratic Republic of Congo. This was an observational study conducted in six study sites where time to death was an outcome of interest. In order to determine factors associated with death they used Cox regression model. The study concludes that Advanced age, history of comorbidity, low level of ambient air SpO2 at admission and long duration of the disease before hospitalization were predictors of survival of patients hospitalized for Covid-19. This is an important study that would support formulation of policy based on local evidence. Generally, the authors attempt to present the paper in a simplistic style. This led to loss of important details. Additionally, there seems to be inconsistency in the arrangement of their articles, making it less succinct. Below are specific comments for each section

Abstract

1. The background section of the abstract jumps to the relevance/significance of the study. Can the authors provide some introduction leading to this work rather than jumping straight to relevance/significance?

2. In methods section of the abstract, the authors indicate that the study was done from 1 January to 31 December 2021. It doesn’t indicate when the study finished for the current analyses.

Introduction

3. The introduction provides rich information on Covid-19. However, the text flow seems inconsistent: It would be important if the authors may re-arrange their argument based on the key themes e.g. background information, what has already been done, the existing gap and the significance/rationale of the study. As it stands, it is hard to read through the introduction and know the messages that the authors are intending to deliver

4. The paragraphs capturing same concepts seem to be unnecessarily split. For example, it can be observed that paragraphs 3 and 4 capture the same concept. The authors should consider rearranging the paragraphs so that they succinctly capture the concepts without unnecessary breaks.

5. In the first paragraph can the authors provide references for the facts presented about covid-19

6. In paragraph 3, can the authors consider correcting the inconsistency on how the dates are reported. This correction should also to the other sections within the manuscript.

Methods

7. In paragraph 3 of the methods section, can the authors provide the rationale for dichotomizing length of stay in the hospital using a cut-off of 5 days.

8. The data collection section seems too brief. Can the authors consider adding detailed information e.g. Who collected the data? How did you ensure data quality considering that you used excel that doesn`t provide trails for any changes? How was the validation and verification done? What version of Stata was used for the analysis? How was the transformation done for each respective variable?

9. The statistical analysis seems too brief. Can the authors consider adding more details e.g. What descriptive statistics where computed and how they were reported, the definition of censoring or loss to follow-up, variable selection procedure/criteria, model selection and diagnostic tests etc.

10. Under the ethical considerations is ESP/KINSHASA ethic review committee? If no, which ethic review committee approved the study?

11. Under the ethical considerations, the authors should consider formatting their text so that it is consistent.

12. Results section

A. In the results section, can the authors consider reporting the total person time, indicate with the text which variables were adjusted for, consider reporting age median(IQR): it can be observed here that the age is skewed.

B. In the results section, can the symptoms be further collapsed based on body organs e.g. digestive disorders (ie. Gastrointestinal disorders), neuro-system disorders (e.g headache), renal disorders, respiratory system disorders, co-morbidity etc.

C. In all the tables decimal should be . not ,

D. The P-Value can not be reported as 0, 000. If it is 0.000 it should be reported as <0.001

E. The hazard ratios should be reported to not more than 2 decimal places and this should be consistent across the other estimates reported in the manuscript

F. It doesn’t make sense that there was an age-group of patients less than 40 years: this less to sparse estimates. Please revisit this and consider collapsing the age categories further

13. Discussion

The authors provide a rich discussion.

A. However, it would be interesting if the authors can highlight what they think are their study observed different findings from the other similar studies [as indicated in their text].

B. Are there any potentially important potential confounders, not collected in this study that could further be interest in future studies

C. Did you consider how could selection bias could potentially affect the current findings? It seems not fully discussed in the current study. For instance, since this is hospital based-surveillance, it means those patients with poor health seeking behavior could not have been part of this study.

6. PLOS authors have the option to publish the peer review history of their article (what does this mean?). If published, this will include your full peer review and any attached files.

**Do you want your identity to be public for this peer review?** For information about this choice, including consent withdrawal, please see our Privacy Policy.

Reviewer #1: No

Reviewer #2: **Yes: **Noel Patson

---

## [Decision Letter · Decision Letter 1]

18 Sep 2023

PGPH-D-23-00180R1

Predictors of Survival among Inpatients in COVID-19 Treatment Centers in the City of Butembo, North Kivu, Democratic Republic of Congo

Dear Dr. Akilimali,

Thank you for submitting your manuscript to PLOS Global Public Health. After careful consideration, we feel that it has merit but does not fully meet PLOS Global Public Health’s publication criteria as it currently stands. Therefore, we invite you to submit a revised version of the manuscript that addresses the points raised during the review process.

We look forward to receiving your revised manuscript.

Kind regards,

Humayun Kabir

Academic Editor

Journal Requirements:

Additional Editor Comments (if provided):

Reviewers' comments:

Reviewer's Responses to Questions

**Comments to the Author**

1. If the authors have adequately addressed your comments raised in a previous round of review and you feel that this manuscript is now acceptable for publication, you may indicate that here to bypass the “Comments to the Author” section, enter your conflict of interest statement in the “Confidential to Editor” section, and submit your "Accept" recommendation.

Reviewer #1: (No Response)

Reviewer #3: (No Response)

Reviewer #4: (No Response)

2. Does this manuscript meet PLOS Global Public Health’s publication criteria? Is the manuscript technically sound, and do the data support the conclusions? The manuscript must describe methodologically and ethically rigorous research with conclusions that are appropriately drawn based on the data presented.

Reviewer #1: Yes

Reviewer #3: (No Response)

Reviewer #4: (No Response)

3. Has the statistical analysis been performed appropriately and rigorously?

Reviewer #1: Yes

Reviewer #3: (No Response)

Reviewer #4: (No Response)

4. Have the authors made all data underlying the findings in their manuscript fully available (please refer to the Data Availability Statement at the start of the manuscript PDF file)?

Reviewer #1: Yes

Reviewer #3: (No Response)

Reviewer #4: (No Response)

5. Is the manuscript presented in an intelligible fashion and written in standard English?

Reviewer #1: Yes

Reviewer #3: (No Response)

Reviewer #4: (No Response)

6. Review Comments to the Author

Reviewer #1: The authors have addressed majority of previous comments, and only minor issues need to be addressed.

Reviewer #3: Use the word Covid-19 as COVID-19.

Please report the number of populations in the abstract.

Please follow reporting guideline such as STOBE

It is needed to clarify how the COVID-19 severity was classified. Provide more details about these measures.

Specify what is the exposure variable.

Provide clear explanation of the time variable of how this was calculated.

Provide the ethical approval number.

Justification of this study is not clearly presented.

Introduction should be more informative at this stage of covid that how this study can add new.

Provide the limitation of the study.

Separate the conclusion from recommendation.

Add section on what new in this study.

Add section for future research directions.

Reviewer #4: It is multivariate or multi-variables?

Please specify if this is which model was used, semi parametric cox proportional hazard model or parametric proportional Cox regression.

The assumption of the semi parametric cox proportional hazard is to be met the proportionality of the hazard.

The assumption of the parametric cox regression is to be met the proportionality of the hazard as well as the distribution of the time variable.

So, specify which model was used in this study.

Do the test for all the assumptions to be met that can be more than one test. For example, for categorical data the test may be not same as for continuous.

Report the graphical presentation of the assumptions. It can be reported as supplementary file so that reader can assess the quality of the assumption as well as the model.

What was done while the proportionality was not met. As I can see in the mortality the KM curve overlap.

Report how the confounding and interaction among the co-variate was addressed.

Report the model fitness such as dfbeta, devodemce residuals, devicence, or Cox-snell residuals. Please provide everything related to the model fitness in the supplementary file.

I want to see that all the comments were carefully addressed to be published this paper at this round of revision.

7. PLOS authors have the option to publish the peer review history of their article (what does this mean?). If published, this will include your full peer review and any attached files.

**Do you want your identity to be public for this peer review?** For information about this choice, including consent withdrawal, please see our Privacy Policy.

Reviewer #1: No

Reviewer #3: No

Reviewer #4: No

---

## [Decision Letter · Decision Letter 2]

13 Dec 2023

Predictors of Survival among Inpatients in COVID-19 Treatment Centers in the City of Butembo, North Kivu, Democratic Republic of Congo

PGPH-D-23-00180R2

Dear Prof Akilimali,

We are pleased to inform you that your manuscript 'Predictors of Survival among Inpatients in COVID-19 Treatment Centers in the City of Butembo, North Kivu, Democratic Republic of Congo' has been provisionally accepted for publication in PLOS Global Public Health.

Best regards,

Julio Croda, Ph.D, M.D.

Academic Editor

Reviewer Comments (if any, and for reference):

Reviewer's Responses to Questions

**Comments to the Author**

1. If the authors have adequately addressed your comments raised in a previous round of review and you feel that this manuscript is now acceptable for publication, you may indicate that here to bypass the “Comments to the Author” section, enter your conflict of interest statement in the “Confidential to Editor” section, and submit your "Accept" recommendation.

Reviewer #4: All comments have been addressed

2. Does this manuscript meet PLOS Global Public Health’s publication criteria? Is the manuscript technically sound, and do the data support the conclusions? The manuscript must describe methodologically and ethically rigorous research with conclusions that are appropriately drawn based on the data presented.

Reviewer #4: Yes

3. Has the statistical analysis been performed appropriately and rigorously?

Reviewer #4: Yes

4. Have the authors made all data underlying the findings in their manuscript fully available (please refer to the Data Availability Statement at the start of the manuscript PDF file)?

Reviewer #4: Yes

5. Is the manuscript presented in an intelligible fashion and written in standard English?

Reviewer #4: Yes

6. Review Comments to the Author

Reviewer #4: No comments this time

7. PLOS authors have the option to publish the peer review history of their article (what does this mean?). If published, this will include your full peer review and any attached files.

**Do you want your identity to be public for this peer review?** For information about this choice, including consent withdrawal, please see our Privacy Policy.

Reviewer #4: No
